# Long-Term Pentoxifylline Therapy Is Associated with a Reduced Risk of Atherosclerotic Cardiovascular Disease by Inhibiting Oxidative Stress and Cell Apoptosis in Diabetic Kidney Disease Patients

**DOI:** 10.3390/antiox13121471

**Published:** 2024-11-29

**Authors:** Jie-Sian Wang, Ping-Hsuan Tsai, Kuo-Feng Tseng, Cheng-Li Lin, Fang-Yu Chen, Chiz-Tzung Chang, Ming-Yi Shen

**Affiliations:** 1Graduate Institute of Biomedical Sciences, China Medical University, 91, Hsueh-Shih Rd., Taichung 40402, Taiwan; 029745@tool.caaumed.org.tw (J.-S.W.); u105010312@cmu.edu.tw (P.-H.T.); u107010409@cmu.edu.tw (K.-F.T.); fyc0321@gmail.com (F.-Y.C.); 2Division of Nephrology, Department of Internal Medicine, China Medical University Hospital, No. 2, Yude Rd., North Dist., Taichung 404327, Taiwan; 019863@tool.caaumed.org.tw; 3Management Office for Health Data, China Medical University Hospital, No. 2, Yude Rd., North Dist., Taichung 404327, Taiwan; orangechengli@gmail.com; 4Department of Medical Research, China Medical University Hospital, No. 2, Yude Rd., North Dist., Taichung 404327, Taiwan; 5Department of Nursing, Asia University, 500, Lioufeng Rd., Wufeng, Taichung 41354, Taiwan

**Keywords:** atherosclerotic cardiovascular disease, diabetes mellitus, chronic kidney disease, pentoxifylline, apoptosis

## Abstract

There is limited understanding of the optimal duration and dosage of pentoxifylline (PTX) therapy required to achieve significant reductions in atherosclerotic cardiovascular disease (ASCVD) risk, particularly in patients with diabetic kidney disease (DKD). This study aimed to evaluate the impact of long-term PTX therapy on the risk of ASCVD in patients with DKD who do not have pre-existing cardiovascular disease, while also exploring potential vascular protective mechanisms. This retrospective cohort study included data from Taiwan’s Ministry of Health and Welfare’s Health and Welfare Data Science Center. In 2008–2019, we identified and analyzed a specific sample of 129,764 patients with DKD without established cardiovascular disease. Participants were categorized according to their PTX treatment regimen. Short-term PTX users (<763 days) had a greater risk of developing ASCVD than non-PTX users. However, those who used PTX for >763 days (long-term PTX treatment) had a significantly lower risk of ASCVD, with a 47% lower cumulative incidence. A dose-dependent reduction in apoptosis was observed via Klotho treatment in cultured human aortic endothelial cells following PTX treatment. Long-term PTX treatment (24 h) caused a higher reduction in H_2_O_2_-induced reactive oxygen species production and cell apoptosis than short-term PTX treatment (2 h). In the DKD mice model experiments, PTX reduced the ASCVD risk by increasing the Klotho levels to inhibit endothelial cell damage. These findings suggest that the cardiovascular and renoprotective benefits of PTX may be extended to primary prevention strategies for people with DKD.

## 1. Introduction

Atherosclerotic cardiovascular disease (ASCVD) is more common in individuals with diabetes mellitus (DM) and chronic kidney disease (CKD) [1]. Diabetic kidney disease (DKD) is also associated with ASCVD [2,3]. The burden of ASCVD has increased substantially globally, with causing 19 million fatalities per year [4,5,6,7].

Moreover, oxidized low-density lipoprotein (Ox-LDL) concentration is associated with ASCVD risk [5,8]. Particularly, the oxidation of the lipid constituents of low-density lipoprotein comprises reactive oxygen species (ROS), which are preferentially involved in this process [9]. Moreover, ROS-induced endothelial cell (EC) apoptosis via cleaved caspase-3 (CC3) may contribute to renal injury and atherosclerosis progression in CKD [1]. Therefore, interventions that suppress ROS-induced EC apoptosis may benefit patients with CKD and cardiovascular disease (CVD) [1].

Klotho is a critical regulator of vascular and renal health, known for its anti-aging and anti-apoptotic properties, and protective effects on ECs [10,11]. In DKD, Klotho levels are often reduced, which is associated with increased oxidative stress, endothelial dysfunction, and accelerated progression of both renal and cardiovascular disease [12]. Low Klotho levels contribute to the upregulation of reactive oxygen species (ROS), exacerbating endothelial damage and promoting atherosclerosis. By increasing Klotho expression, PTX may help reduce ROS production, protect endothelial function, and thereby lower ASCVD risk. This mechanistic pathway provides a potential link between PTX therapy and improved cardiovascular outcomes in DKD patients [10,11,13]. Pentoxifylline (PTX) inhibits cytokines and reduces ROS generation via granulocytes and ECs and decreases the development of hypercholesterolemic atherosclerosis [14]. Furthermore, in vivo and in vitro studies have examined the suboptimal use of PTX and the achievement of Ox-LDL goals [15,16]. Additionally, PTX improves the estimated glomerular filtration rate and proteinuria [17,18,19]. PTX’s anti-inflammatory properties may be particularly beneficial for DKD patients, as they often experience chronic inflammation that exacerbates both renal and cardiovascular disease [20,21]. Its renoprotective effects can help preserve kidney function, which is crucial for reducing the ASCVD risk in this vulnerable population. While other therapeutic agents, such as statins and ACE inhibitors, are commonly used for ASCVD prevention [22], they may pose risks of adverse effects in DKD patients, particularly concerning renal function [23]. In contrast, PTX offers a unique approach with its dual anti-inflammatory and renoprotective properties, making it a potentially safer option for this population. Therefore, examining PTX therapy can reduce the increased burden of ASCVD worldwide [24]. In addition, the rationale for comparing long-term versus short-term PTX therapy lies in the progressive nature of both DKD and ASCVD, which are chronic conditions requiring sustained management. While PTX’s short-term anti-inflammatory and renoprotective effects are well documented, it remains unclear whether these benefits accumulate over extended periods to provide more substantial protection against cardiovascular events. Given the chronicity of both diseases, understanding whether prolonged PTX therapy offers enhanced or sustained benefits is critical for optimizing the treatment strategies for managing DKD patients at risk of ASCVD.

However, little is known about which PTX regimen provides optimal ASCVD outcomes. Accordingly, we aimed to confirm the protective function and mechanism of PTX in vascular ECs and examine whether long-term PTX therapy might be better than short-term therapy in reducing the incidence of ASCVD in Taiwanese patients with DKD. The dual clinical and mechanistic effects of PTX, particularly its ability to reduce inflammation, oxidative stress, and endothelial damage through mechanisms such as Klotho upregulation and ROS reduction, highlight its potential as a more targeted therapy for DKD patients at risk of ASCVD. By elucidating these pathways, this study provides a foundation for more tailored treatments aimed at specific molecular targets, such as enhancing Klotho expression or mitigating ROS-mediated vascular damage. Future treatment strategies could leverage these mechanistic insights to refine PTX dosing, optimize therapy duration, or combine PTX with other agents to further enhance its protective effects, leading to improved cardiovascular outcomes in this high-risk population.

## 2. Materials and Methods

### 2.1. Data and Sample Source

The National Health Insurance Research Database (NHIRD) in Taiwan is a prime example of a population-level data source that may be used to produce empirical evidence to support clinical judgments and health care policy decisions [25]. To confirm the protective function and mechanism of PTX in vascular ECs and examine whether long-term PTX therapy might be better than short-term therapy in reducing the incidence of ASCVD in Taiwanese patients with DKD, this study used data from the Health and Welfare Data Science Center of Taiwan’s Ministry of Health and Welfare, including the NHIRD and the Registry for Catastrophic Illnesses and Death Records. All research endeavors adhered to the guiding principles of the Declaration of Helsinki’s. Because the study was retrospective in nature, informed consent was not required. The study design was authorized by the Institutional Ethics Review Board of China Medical University Hospital (No. CMUH110-REC1-038).

### 2.2. Study Participants

Established in 1995 as a universal, mandatory health insurance scheme, the Taiwan National Health Insurance (NHI) program covered more than 98% of Taiwan’s 23 million residents as of 1998. The NHI Research Database (NHIRD), created by the Taiwanese government, contains the claims information of Taiwan’s NHI policyholders. In the NHIRD between January 2008 and December 2019, we selected 535,010 patients with newly diagnosed DM (ICD-9: 250/ICD-10: E08-E13) and CKD (International Classification of Diseases (ICD)-9: 585/ICD-10: N189). Next, we excluded patients aged <20 years (*n* = 1010) and those with a history of DM or CKD for ˂90 d (*n* = 74), ASCVD at baseline (*n* = 117,237), ASCVD outside the study period (*n* = 16), and those without sex information (*n* = 428). Overall, we selected 129,764 patients who were subcategorized into two PTX sub-cohorts. PTX treatment was initiated on the index date (day 91). We randomly selected a non-PTX cohort with a propensity score-matched sample size similar to that of the PTX cohort. Additionally, a multivariate logistic regression model was used to calculate a propensity score based on the 5-year age group, sex, year of diagnosis of DM and CKD, and comorbidities. DKD diagnosis was defined based on the presence of both diabetes mellitus and chronic kidney disease (CKD). CKD was diagnosed according to the following criteria: an estimated glomerular filtration rate (eGFR) of less than 60 mL/min/1.73 m^2^ and/or the presence of albuminuria (urinary albumin-to-creatinine ratio ≥ 30 mg/g) for at least three months, in accordance with established clinical guidelines. These criteria were used to ensure accurate identification of DKD patients without pre-existing ASCVD in the cohort. Figure 1 presents a detailed flowchart for establishing the study cohort.

### 2.3. Outcome and Variables

From the index date to the end of 2019, all participants were followed up with until ASCVD occurrence (ICD-9/ICD-10:429.2/I25.10), death, or withdrawal from insurance. The primary outcome was an ASCVD diagnosis, calculated using the 10-year ASCVD risk estimation chart recommended in the 2013 American College of Cardiology Guideline for the Management of Dyslipidemia in Adults [26]. We included acute myocardial infarction (AMI), hypertension (HTN), chronic obstructive pulmonary disease (COPD), liver cirrhosis (LC), and autoimmune diseases as comorbidities possibly associated with the outcomes. Table 1 presents the characteristics of the participants.

### 2.4. In Vitro Cell Culture and Assay for Cellular Viability and Apoptosis

The primary human aorta ECs (HAECs) in an EGM2 medium (Lonza, Basel, Switzerland) used in our cell investigations are healthy, disease-free human aortic ECs [1]. A 12-well plate of HAECs was cultured until it reached 80% confluence before being used for the cell experiments. Briefly, 4 × 10^3^ cells were seeded into 96-well plates to determine their viability. Next, the cells were incubated with PTX (0–500 μM) for 24 h to perform the toxicity assays. A 24 h incubation period with PTX (6.25–50 μM) or the control vehicle (0.1% dimethylsulfoxide [DMSO]) was followed by a 24 h incubation period with or without H_2_O_2_ (100 μM). Afterward, each well was treated with 20 mL of 3-(4,5-dimethylthiazol-2-yl)-2,5-diphenyltetrazolium bromide solution (5 mg/mL) for 4 h at 37 °C. Then, DMSO was substituted for the supernatant, and oscillations were observed.

An infinite M1000 microplate reader (Tecan Group AG, Männedorf, Switzerland) was used to measure the absorbance at 570 nm. To determine the percentage of the control, we normalized all data to the corresponding controls. Hoechst 33342 (1 mM) and calcein acetoxymethyl ester (calcein-AM) (Molecular Probes, Eugene, OR, USA) were used to quantify apoptosis. Next, an inverted microscope (IX70; Olympus, Tokyo, Japan) was used to quantify apoptotic cells as previously described [27].

### 2.5. Measurement of Intracellular ROS Levels

An assay kit (Abcam, Cambridge, UK) was used to measure the amounts of cellular ROS and superoxide following the manufacturer’s recommendations. Initially, 1.0 × 10^4^ cells/well were added to 96-well plates, and the cells were left to attach for 24 h. Following the removal of the culture supernatant, the cells were cleaned in phosphate-buffered saline (PBS). Next, a ROS-specific dye called 2′,7′-dichlorofluorescein diacetate (DCFH2-DA) was added, and the mixture was incubated for 30 min in the dark at 37 °C. After being incubated, the cells were cleaned in PBS twice. Finally, we employed fluorescence microscopy to analyze cells using the Infinite M1000 microtiter plate reader (Tecan Group AG) (Ex = 488 nm, Em = 520 nm) to detect intracellular ROS levels.

### 2.6. Forecasting Interactions Between Chemicals and Proteins

Databases and online resources, such as the Search Tool for Interactions of Chemicals (STITCH) (version 5), were utilized to forecast protein–protein and chemical–protein interactions [28]. To explore the effect of PTX on renal disease-related pathways, depending on the color saturation of the edges, we determined the level of confidence in the functional association.

### 2.7. Protein Extraction and Western Blotting

In 6-well plates, 2 × 10^5^ cells/well were planted, and the cells were left to attach for 24 h. We used the vehicle, H_2_O_2_ (100 μM), for 24 h in isolation, or H_2_O_2_ (100 μM) following a 24 h pretreatment with PTX. Using a lysis buffer containing protease inhibitors, we homogenized the cells and lysed them for protein extraction (Roche Applied Science, Penzberg, Germany). Protein concentrations were ascertained using a bovine serum albumin test kit (Pierce, Waltham, MA, USA). After loading onto a 12% SDS polyacrylamide gel, the cell lysates (containing 20 μg of protein) were separated using a sodium dodecyl sulfate (SDS) polyacrylamide gel electrophoresis technique. Following protein separation, Super Block was used to block proteins before they were placed onto Hybond-PVDF membranes (GE Healthcare Amersham, Buckinghamshire, UK). Next, 5% non-fat dry milk in PBS-Tween-20 (PBST) was used to block non-specific sites for 60 min. For immunoblotting, one of the following primary antibodies was employed at a 1:1000 dilution: anti-β actin (1:10,000; Sigma, Cheshire, UK), anti-cleaved caspase-3 (Cell Signaling), and anti-Klotho (Cell Signaling, Danvers, MA, USA).

As the secondary antibody, anti-rabbit horseradish peroxidase-conjugated immunoglobulin G (1:1000; DakoCytomation, Glostrup, Denmark) in PBST was used for 60 min at room temperature. In order to verify uniform protein loading in every lane, we employed α-actin. Next, we used ECL reagents (Millipore, St. Louis, MO, USA) and Quantitative Video Densitometry (G-Box Image System; Syngene, Frederick, MD, USA) to quantify protein expression [29].

### 2.8. SiRNA Transfection

In six-well plates, roughly 5 × 10^6^ cells were inoculated each well. Fresh medium enriched with 20% FBS but devoid of antibiotics was used in place of the old medium. As directed by the manufacturer, control- and Klotho-siRNA (GeneDireX, Inc., Las Vegas, NV, USA) were made at a final concentration of 10 nM. After 10 min of mixing the control- or Klotho-siRNA with the transfection reagent, the cells were transfected for 12 h; then, H_2_O_2_ was added to activate the cells.

### 2.9. Quantitative Polymerase Chain Reaction in Real Time

Isolated total RNA was extracted from frozen materials at −80 °C using NucleoZOL, according to manufacturer’s instructions. Complementary DNA (cDNA) was created using the iScript cDNA Synthesis Kit (Bio-Rad, Hercules, CA, USA). Lastly, iQTM SYBR Green Supermix (Bio-Rad) was used to perform the real-time polymerase chain reaction (PCR) [1]. The following primers were used: human Klotho (forward) 5′-ACTCCCCCAGTCAGGTGGCGGT A-3′ and human Klotho (reverse) 5′-TGGGCCCGGGAAACCATTGCTGTC-3′; mouse Klotho (forward) 5′-GATGGCAGAGAAATCAACACAGT-3′ and mouse Klotho (reverse) 5′-ACTACGTTCAAGTGGACACT-3′.

### 2.10. Experimental Animals and Study Design

The recommendations of the “Guide for the Care and Use of Laboratory Animals” published by the United States National Institutes of Health (NIH Publication No. 85-23, revised 1996) for the feeding of animals, related surgery, and collection of blood and tissue samples were followed as close as possible. All animal experiments and implementation procedures were approved by the China Medical University Laboratory Animal Care and Use Committee (approval number: CMUIACUC-2020-108). Isoflurane inhalation anesthesia was used to perform subtotal nephrectomy (5/6 Nx) on 10–12-week-old streptozotocin (STZ) mice. The right kidney was completely removed, and two or three renal arteries were ligated to selectively infarct approximately two-thirds of the left kidney. The mice received PTX (50 mg/kg) for 8 weeks (SD) or 12 weeks (LD) after 5/6 Nx surgery. Finally, all mice were euthanized via isoflurane inhalation, followed by cervical dislocation [1,30,31]. The doses of PTX used in the cell and animal studies were selected based on previous preclinical research, which demonstrated effective anti-inflammatory and renoprotective effects at these concentrations. In the animal model, the dose was adjusted for body surface area to approximate the therapeutic dose typically administered to humans, as per standard dose conversion guidelines between species [32]. Although establishing the exact dose equivalency between humans and animals is challenging, the chosen doses are within the range considered pharmacologically relevant to PTX’s clinical use in humans for managing inflammatory and cardiovascular conditions.

### 2.11. Assay for Lipid Peroxidation, Oil Red O Staining, and Histological Staining of Aortic Sections

The plasma, aorta, and kidneys were collected after euthanasia. A malondialdehyde (MDA) assay kit (No. ab118970; Abcam) was used to assess lipid peroxidation. To visualize atherosclerotic plaques, we performed Oil Red O staining 1 h (Sigma) [33]; digital photographs were taken using a Canon EOS 70D digital camera (Tokyo, Japan). An overnight fixation with 4% paraformaldehyde was followed by paraffin embedding of the aortic roots. Slices of 3 μm were cut from serial sections. Hematoxylin and eosin (H&E) staining was used to every third slide from the serial sections, and images were acquired using a Leica DM750 (Wetzlar, Germany).

### 2.12. Statistical Analyses

Numbers (percentages) are used to communicate categorical variables; means and standard deviations (SD) are used to express continuous variables. Categorical and continuous variables were compared using chi-square and *t*-tests, respectively. Medians (first and third quartiles) were presented for non-normally distributed continuous variables, and Wilcoxon rank-sum tests were used to compare the two groups. Based on the number of incident cases divided by the number of follow-ups (person-years), we calculated the ASCVD incidence rate (IR). The adjusted hazard ratios (aHRs) and 95% confidence intervals (CIs) for ASCVD were computed using Cox proportional hazards regression analysis. A Kaplan–Meier analysis was also used to examine differences in ASCVD cumulative incidence rates between both cohorts. In the statistical analysis, we adjusted for a range of potential confounders known to influence ASCVD risk and outcomes in DKD patients. The variables included in the adjustment across different models were age, sex, body mass index (BMI), smoking status, blood pressure, lipid levels, baseline renal function (eGFR), albuminuria, diabetes duration, glycemic control (HbA1c), use of antihypertensive and lipid-lowering medications, and comorbidities such as hypertension and dyslipidemia. These variables were selected based on their established associations with both DKD and ASCVD risk. Adjusted variables were defined as the covariates that were included in our statistical models to control for potential confounding factors when assessing the effects of PTX on the cardiovascular outcomes.

## 3. Results

We established the PTX and non-PTX cohorts, each comprising 31,141 patients with DM and CKD. The distributions of age, sex, and comorbidities were similar between the PTX and non-PTX cohorts. The non-PTX and PTX cohorts had mean ± SD follow-up periods of 3.09 ± 2.13 and 3.18 ± 2.20 years, respectively. Table 1 summarizes the characteristics of each group.

### 3.1. Incidence and Hazard Ratios of ASCVD

Table 2 shows the incidence and hazard ratios of ASCVD after 12 years of follow-up. Quartile-based time points were used to categorize the duration of dosing administration as follows: 1–70, 71–308, 309–763, and >763 d. Table 3 shows the proportion of different PTX usage and the distribution of days-supply in the participants with DM and CKD. In the cohort study, long-term PTX therapy (defined as >763 days of use) was associated with a 47% reduction in ASCVD incidence compared to short-term PTX use. Specifically, the hazard ratio (HR) for ASCVD in long-term PTX users was 0.53 (95% CI: 0.49–0.58, *p* < 0.001), indicating a statistically significant protective effect. These results highlight the potential long-term benefits of PTX in reducing cardiovascular risk among DKD patients.

### 3.2. Cumulative ASCVD Incidence

When the cumulative incidence of ASCVD was simulated, participants using PTX for 1–70 d could not achieve their ASCVD goals. Over 99% of the participants using PTX for 71–308 d could not also achieve their ASCVD goals. However, when participants used PTX for 309–763 d, they promptly achieved their ASCVD goals but failed to achieve them in the following years. The corresponding ASCVD goals were achieved by all participants who received PTX treatment for >763 d (Figure 2). The term “ASCVD goals” refers to achieving clinically recommended targets for reducing the risk of ASCVD, such as lowering LDL cholesterol, managing blood pressure, and controlling blood glucose levels in DKD patients. In our study, some patients receiving PTX therapy did not reach these targets during the follow-up period, particularly in the short-term PTX group. Specifically, patients in this group showed only modest improvements in LDL reduction and glycemic control, which may have contributed to the continued elevated risk of ASCVD over the following years.

### 3.3. Pentoxifylline Inhibits H_2_O_2_-Induced Cellular Apoptosis in HAECs

Pentoxifylline (250 μM) did not show any significant cytotoxic effects on HAECs, as determined using the water-soluble tetrazolium salt (WST-1) assay (Figure 3A). Moreover, the cells were pre-incubated with various PTX concentrations (6.25–25 μM) for 24 h and subsequently challenged with H_2_O_2_ (100 μM) for 24 h. The viability of HAECs was significantly reduced after H_2_O_2_ (100 μM) treatment compared with the control (*p* < 0.001, *n* = 3), whereas pretreatment with PTX (6.25–25 μM) followed by H_2_O_2_ (100 μM) treatment increased the cell viability in H_2_O_2_-treated HAECs (Figure 3B). The ROS assay was performed using DCFH-DA staining. H_2_O_2_ significantly increased the level of ROS production, peaking after 2 h, compared with that in the control group, and PTX (6.25–25 μM) inhibited this effect in a concentration-dependent manner (Figure 3C). These findings suggested that PTX might arrest H_2_O_2_-induced intracellular ROS generation.

We used the Hoechst 33,342 fluorescence stain (nuclear morphology) and Calcein-AM (membrane integrity) to stain treated cells to assess whether PTX shields HAECs against H_2_O_2_-induced cell damage via apoptosis. We regarded HAECs with fragmented, condensed nuclei as apoptotic. H_2_O_2_ alone caused apoptosis (*p* < 0.001), according to fluorescence microscopy, and PTX (6.25–25 μM) pretreatment reduced the H_2_O_2_-induced apoptosis in a dose-dependent manner (Figure 3D,E). Furthermore, we explored the effects of PTX on renal disease-related pathways and predicted protein–protein and chemical–protein interactions using databases and web resources, namely, STITCH v5 [28].

The STITCH database integrates experimental and manually curated evidence with text mining and interaction predictions to provide a complete characterization of protein–chemical interactions [34]. PTX facilitates protein–protein interactions between genes, and the line size was proportional to the combined fraction of interactions. The results also showed that Klotho is the major protein associated with PTX. Furthermore, PTX, ROS (H_2_O_2_), Klotho, and caspase 3 (CASP3, an apoptosis-associated protein) were closely associated with renal disease-related pathways (Figure 3F).

The effect of PTX treatment on Klotho expression was determined via real-time PCR. Klotho levels were significantly reduced by H_2_O_2_ treatment (Figure 3G, *p* < 0.05, *n* = 3). However, they were increased in PTX (6.25–25 μM)-pretreated, H_2_O_2_-induced HAECs (Figure 3G). Additionally, the protein expression levels of Klotho and CC3 were determined using Western blotting. H_2_O_2_ treatment resulted in a significant increase in CC3 and reduction in Klotho (Figure 3H), whereas CC3 levels were reduced and Klotho increased in the PTX (6.25–25 μM)-pretreated, H_2_O_2_-induced HAECs (Figure 3H). Next, we investigated the role of Klotho in the protective effects of PTX against H_2_O_2_-induced increased intracellular ROS level and decreased cell viability in human aortic endothelial cells (HAECs). HAECs were transfected with either a scrambled control siRNA (si-Ctrl) or siRNA targeting Klotho (si-Klotho) for 12 h, followed by pretreatment with PTX (25 µM) and subsequent stimulation with H_2_O_2_ for an additional 24 h.

Our results demonstrated that PTX pretreatment significantly reduced intracellular ROS level and increased cell viability in H_2_O_2_-stimulated HAECs. However, knockdown of Klotho with si-Klotho abrogated these effects, leading to increased intracellular ROS level and reduced cell viability despite PTX treatment. In contrast, transfection with si-Ctrl did not interfere with the inhibitory effects of PTX on H_2_O_2_-induced increased intracellular ROS level and reduced cell viability (Figure 3I,J).

These findings strongly support the hypothesis that PTX attenuates endothelial cell apoptosis through a Klotho-dependent mechanism.

### 3.4. Protective Effect of Short- or Long-Term Pentoxifylline Exposure Against H_2_O_2_-Induced ECs

In WST-1 and apoptosis assays, we examined the protective effects of PTX exposure at several time points (2 h and 24 h) against H_2_O_2_-induced EC damage. H_2_O_2_ (100 μM)-induced HAECs pretreated with PTX for 24 h showed significantly increased cell viability (Figure 4A, *p* < 0.001, *n* = 4), but this was not the case in those pretreated with PTX for 2 h. The intracellular ROS levels in H_2_O_2_ (100 μM)-induced cells treated with PTX for 24 h were significantly reduced, whereas PTX pretreatment for 2 h did not have the same effect (Figure 4B).

In the apoptosis assay, cells pretreated with PTX for different intervals (2 h and 24 h) showed a time-dependent significant inhibition of H_2_O_2_ (100 μM)-induced cell damage (Figure 4C,D). The mRNA (Figure 4E) and protein (Figure 4F) levels of Klotho in HAECs increased in cells pretreated with PTX for 24 h but not in those pretreated for 2 h. These findings show that long-term PTX pretreatment may provide better inhibitory effects against H_2_O_2_-induced cell damage than short-term PTX pretreatment.

### 3.5. PTX Inhibits Atherosclerosis in a DKD Mouse Model

Figure 5A shows a representation of the mouse model for DKD. The mice showed significantly elevated lipid accumulation and atherosclerosis lesion size compared with those in a mouse model for DM (STZ-induced C57BL/6 mice) and sham-operated wild type (WT) mice (Figure 5B,C, *p* < 0.01). Long-term PTX treatment (LD) significantly reduced lipid accumulation (Figure 5B,C) compared with short-term PTX treatment (SD) in DKD mice. The ROS levels were higher in the aorta, kidney, and plasma of DKD mice than in those in DM and sham mice (Figure 5D–F). Moreover, long-term treatment with PTX significantly reduced ROS levels in DKD mice (Figure 5D–F), but short-term treatment did not.

Histopathologic analyses of the aortas with H&E staining and terminal deoxynucleotidyl transferase-mediated dUTP nick end labeling (TUNEL) were performed to assess the thickness of the aorta and apoptosis levels. Apoptosis and aorta thickness were significantly increased in the DKD mice compared with in the DM and sham mice (Figure 5G–I). However, aorta thickness was significantly reduced in the DKD mice with the long-term PTX treatment, as revealed by the results of H&E staining (Figure 5G,H) compared with those subjected to short-term PTX treatment.

The mRNA and protein levels of Klotho were decreased, and CC3 protein expression was increased in the DKD mice compared with the DM and sham mice. However, DKD mice treated with PTX for a long term showed a significant increase in the mRNA and protein levels of Klotho, and a reduced CC3 protein expression (Figure 5J,K) compared with those treated with PTX for a short term. Overall, the in vivo findings support our in vitro results, indicating that long-term PTX treatment may have better inhibitory effects on ROS-induced EC apoptosis and atherosclerosis than short-term PTX treatment in DKD mice.

## 4. Discussion

The primary objective of our study was to assess the differential effects of long-term versus short-term PTX therapy on ASCVD risk in DKD patients and to elucidate the underlying cellular mechanisms that might explain these effects. Our findings demonstrate a significant reduction in ASCVD risk with long-term PTX therapy (>763 days), but not with short-term therapy. This result aligns with the observed mechanistic benefits of PTX in experimental models, where long-term exposure resulted in sustained reductions in ROS production, apoptosis, and endothelial dysfunction, particularly through the upregulation of Klotho. These cellular mechanisms likely contribute to the delayed but cumulative cardiovascular protective effects observed in the clinical cohort over time. The distinction between long-term and short-term therapy is critical, as our results suggest that the full protective benefits of PTX may only become evident with extended treatment durations, supporting the rationale for sustained therapy in high-risk DKD populations.

In this study, we examined whether PTX might be useful in decreasing ASCVD progression in patients with DKD and investigated the optimal duration of PTX treatment. The average follow-up period was 3.18 years. The Kaplan–Meier analysis showed a lower cumulative ASCVD incidence (log-rank test, *p* < 0.001) in the subgroup treated with PTX for more than 763 days than in the subgroup receiving PTX for a short period. Although the risk of ASCVD was higher in patients with DKD receiving PTX for a short period, it was reduced by 47% in PTX users who were treated with the drug for more than 763 days.

The 47% reduction in the ASCVD risk observed among long-term PTX users in our study is consistent with previous findings that highlight PTX’s anti-inflammatory, renoprotective, and endothelial protective effects [21]. However, the long-term efficacy of PTX has not been as widely studied in the context of ASCVD prevention in DKD patients, making our findings a novel contribution to the literature. Prior studies have predominantly focused on PTX’s short-term effects, showing improvements in renal outcomes and inflammation markers, but without extensive follow-up on cardiovascular events. Our results suggest that sustained PTX therapy may be necessary to achieve significant cardiovascular benefits, likely due to the cumulative effects on endothelial health, reduction in ROS, and Klotho upregulation over time.

PTX is an anti-inflammatory and hemorheological drug with favorable effects on kidney function. It is relatively safe to use, with only minor gastrointestinal adverse effects reported [20]. Moreover, PTX therapy is a key strategy for ameliorating proteinuria and reducing renal damage in patients with diabetic nephropathy [20]. In individuals with and without diabetes mellitus and CKD, elevated albuminuria or proteinuria is a strong risk factor for CVD, and the administration of antiproteinuric medications lowers the incidence of cardiovascular events [20]. PTX is a potent antioxidant that inhibits ROS generation by activating antioxidant enzymes [21]. Moreover, ROS plays a critical role in cardiovascular injury induced by kidney damage [1]. When comparing PTX with other therapies commonly used for ASCVD prevention in DKD patients, such as statins, angiotensin-converting enzyme (ACE) inhibitors, and sodium-glucose cotransporter-2 (SGLT2) inhibitors, it is important to note that these agents primarily target lipid levels, blood pressure, and glycemic control. In contrast, PTX exerts its effects through anti-inflammatory and anti-apoptotic pathways, offering a complementary mechanism of action. While statins and SGLT2 inhibitors have demonstrated clear benefits in reducing ASCVD risk, PTX’s unique mechanism—especially its influence on Klotho and ROS—may provide an additional therapeutic benefit, particularly in patients who are refractory to standard treatments. Further comparative studies are warranted to explore the potential synergistic effects of PTX when combined with these therapies.

The major mechanisms contributing to CVD development in patients with DKD include traditional cardiovascular risk factors (hyperglycemia) and DKD-associated mediators (inflammation, proteinuria, and ROS). Moreover, their contribution to ASCVD is particularly important during the early stages of CKD [35]. Moreover, EC damage is critical in CVD progression in patients with DKD; ROS cause EC damage [1]. Additionally, increased levels of ROS and their remnants accelerate DKD and atherosclerosis. In this study, PTX treatment significantly reduced H_2_O_2_-induced EC apoptosis in a dose-dependent manner. Long-term PTX pretreatment had a more pronounced protective effect than short-term pretreatment against apoptosis in H_2_O_2_-induced HAECs. Long-term PTX pretreatment significantly reduced H_2_O_2_-induced EC damage and inhibited intracellular ROS levels induced by H_2_O_2_. Moreover, the acute benefits of PTX on cellular mechanisms (e.g., reducing ROS and apoptosis) may contribute to long-term vascular health; however, these molecular changes may need sustained intervention over time to translate into measurable reductions in the ASCVD risk. Differences in the study design, such as the chronicity of disease in human subjects and timing of clinical events, may also account for this delayed protective effect.

Additionally, Klotho is critical in diabetic atherosclerosis and is causally linked to DKD [36]. Klotho deficiency increases endogenous ROS production and exacerbates oxidative stress [37]. The STITCH database helps to explore known or predicted chemical and protein interactions. In this study, we used databases and web resources (STITCH version 5 [28]) to demonstrate that PTX, ROS (H_2_O_2_), Klotho, and caspase 3 closely interact with renal disease-related pathways. PTX increased the Klotho mRNA and protein levels and decreased the CC3 levels in H_2_O_2_-induced HAECs. These findings suggest that PTX inhibits arterial EC damage via Klotho.

In CKD patients, hyperglycemia is highly correlated with CVD development [38]. Regarding end-stage renal disease and CVD in CKD patients, DKD is the most common cause. Therefore, novel therapies to suppress DKD are required. 

Testing new treatments and understanding the pathophysiology of diseases are two benefits of using rodent models. Progressive albuminuria, a decline in renal function, and the distinctive histological alterations in glomeruli and tubulointerstitial lesions seen in human DKD should all be present in an animal model of the disease [31]. The STZ-induced mouse model of DM is the most common animal model used to study early DKD [31]. Additionally, subtotal 5/6 nephrectomy is the most frequently used rodent CKD model [1]. In this study, we used 5/6 Nx surgery in STZ-induced C57BL/6 mice as an animal model of DKD to confirm our findings. Long-term treatment with PTX was found to be more effective than short-term treatment in mice and cell culture models. Although some data on short-term treatment appear to be beneficial, factors such as disease rebound effect, insufficient drug accumulation, or treatment interruptions affect efficacy and may contribute to an increased risk of CVD in patients treated with short-term PTX. Long-term PTX treatment dramatically reduced apoptosis in the aorta; peroxidation in the kidney, plasma, or aorta; and lipid buildup in DKD mice fed a high-fat diet. These effects were greater than those of short-term PTX treatment.

Vascular inflammation, atherosclerosis, and lipid metabolism are associated with DKD and ASCVD [1,39], and PTX may be effective in improving the management of these diseases. In a 6-month clinical trial, the carotid intima-media thickness decreased in the PTX group and marginally increased in the placebo group among adolescents with type 1 diabetes [40]. The latest study also discovered that PTX had a positive effect linked to Klotho gene expression in peripheral blood cells, reducing subclinical atherosclerosis progression as measured by carotid intima-media thickness variation [12]. In addition, in rabbits administered a diet high in cholesterol, a clinically appropriate daily oral dose of PTX (40 mg/kg) reduced the area of aortic atherosclerotic plaques by 38% [14]. Although the drug did not influence serum lipid levels, MDA levels, which are a measure of oxidative stress in the plasma and aorta, were reduced by 32% and 37%, respectively, in the PTX group relative to the controls [14]. This is likely because plaques tend to contain few neutrophils. Infiltrating monocytes, and possibly ECs, mediate the anti-inflammatory effect [41]. The PTX-treated group was reported to experience significant reductions in serum C-reactive protein and TNF-α compared with the placebo group [41]. Although PTX has been extensively studied, only a few studies have examined its long-term impact on spontaneous stroke and cerebral ischemia [42,43]. Further, standard-dose PTX (1200 mg/d) or aspirin/dipyridamole was administered to the patients with other appropriate medications, and the outcomes assessed included death, nonfatal stroke, and transient ischemic attack. The PTX and the aspirin/dipyridamole groups included 15 (15%) and 29 (27%) patients who achieved this outcome, respectively [42].

The delayed protective effects of PTX on ASCVD risk observed in our human cohort can be better understood by considering the underlying mechanisms elucidated in our in vitro and animal studies. One key pathway involves the reduction in reactive oxygen species (ROS) and upregulation of Klotho, both of which are crucial for maintaining endothelial function. In our animal models, PTX treatment led to a significant reduction in the ROS levels, thereby decreasing endothelial cell apoptosis and oxidative stress, which are central to the pathogenesis of atherosclerosis. Additionally, PTX was shown to increase Klotho expression, a protein known for its anti-aging properties and its protective effects on the vasculature [36]. Klotho enhances nitric oxide production and inhibits oxidative stress, which helps preserve endothelial health. However, these protective effects are not immediate and require sustained treatment to manifest fully, as seen in both our cellular models and the cohort study. In the context of chronic diseases such as DKD, where patients experience ongoing endothelial dysfunction, the cumulative impact of long-term ROS reduction and Klotho upregulation may only become apparent after extended PTX therapy. This could explain why significant cardiovascular benefits were observed only after two years of PTX use in the human cohort. The mechanistic insights from the experimental models provide a biological basis for the delayed but sustained ASCVD risk reduction in humans, as long-term exposure to PTX may be necessary to fully restore endothelial function and confer cardiovascular protection.

Several limitations must be considered when interpreting the results of this study. First, while liver disease and COPD were used as proxies for BMI, alcohol intake, and smoking, these indirect measures may not fully capture the influence of these lifestyle factors on the ASCVD risk. Consequently, the inability to directly adjust for BMI, alcohol consumption, and smoking status may introduce residual confounding, which could impact the observed associations between PTX use and ASCVD risk. Second, the absence of lipid profile data—an essential component for assessing cardiovascular risk—represents a significant limitation. Without detailed information on the LDL, high-density lipoprotein, and triglyceride levels, it is difficult to account for lipid-related cardiovascular risk factors, which are crucial in ASCVD prevention [44]. This limitation may affect the study’s ability to fully evaluate the potential benefits of PTX, particularly in comparison to other lipid-lowering therapies commonly used in DKD patients. Third, the use of claims data, while valuable for large-scale observational studies, lacks the clinical granularity necessary to assess the full range of patient characteristics, such as renal function, glycemic control, or medication adherence, which may influence the outcomes. The reliance on demographic consistency with the census data helps ensure representativeness but does not fully eliminate potential biases associated with incomplete clinical details. Fourth, comparability of treatment durations across species: we recognize the challenge of directly comparing the 12-week treatment duration in mice to over two years in humans As there are differences in pharmacokinetics and physiology between species which can complicate direct extrapolation of the results. While the mechanistic insights gained from the animal study (e.g., reduction in ROS and apoptosis) are relevant to the pathophysiology of atherosclerosis and DKD, we agree that the long-term effects of PTX in humans may not be fully mirrored by the short-term treatment duration in mice. This uncertainty has now been explicitly acknowledged, along with a call for further studies to bridge this gap. Fifth, the cellular responses observed in our study may differ in ECs derived from DKD patients with these specific pathophysiological changes. While PTX has been shown to have protective effects on ECs in various contexts, the extent of its efficacy in cells from DKD patients may vary, which could influence the translatability of our findings. While the cellular and animal studies provide valuable mechanistic insights, we acknowledge that the complexity of human DKD and its associated endothelial dysfunction may require more tailored models to fully understand the long-term effects of PTX in this population. We also suggest that future studies should consider using patient-derived ECs to better capture the disease-specific cellular environment.

## 5. Conclusions

Long-term PTX therapy was associated with a significant 47% reduction in ASCVD risk among DKD patients without pre-existing cardiovascular disease. This finding underscores the potential of PTX as a preventive therapy for high-risk populations. Mechanistic studies further revealed that PTX reduces oxidative stress and endothelial damage through Klotho upregulation, providing a biological basis for its protective effects. While the results are promising, the lack of lipid profile data and the reliance on observational data highlight the need for further prospective studies and clinical trials to confirm the therapeutic role of PTX in ASCVD prevention. Nonetheless, these findings suggest that PTX could offer a novel and complementary approach to current ASCVD prevention strategies in DKD patients, particularly when long-term therapy is implemented.

## Figures and Tables

**Figure 1 antioxidants-13-01471-f001:**
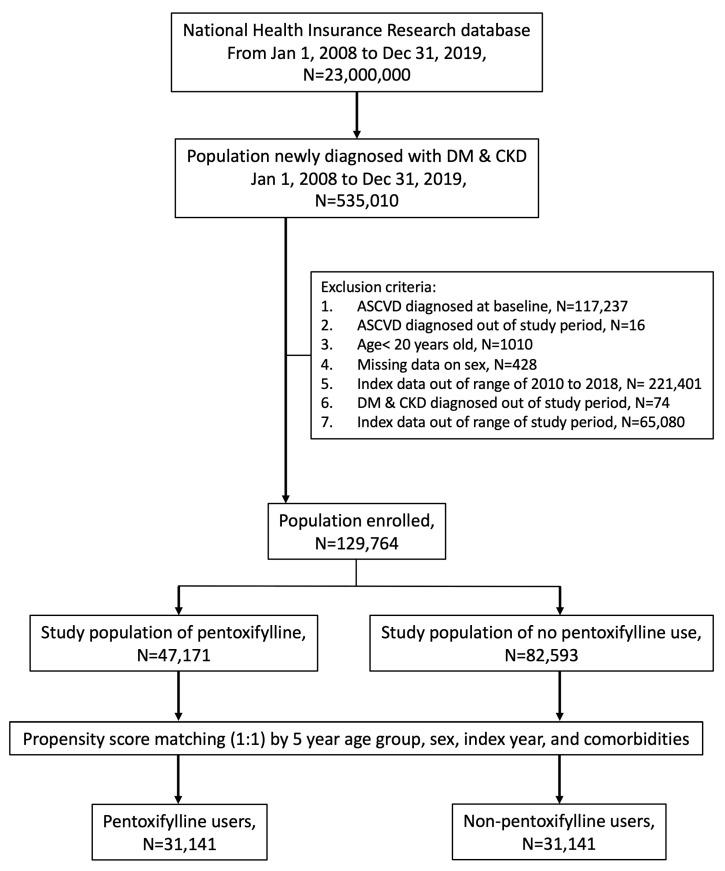
Flowchart for the study population. DM, diabetes mellitus; CKD, chronic kidney disease; ASCVD, atherosclerotic cardiovascular disease.

**Figure 2 antioxidants-13-01471-f002:**
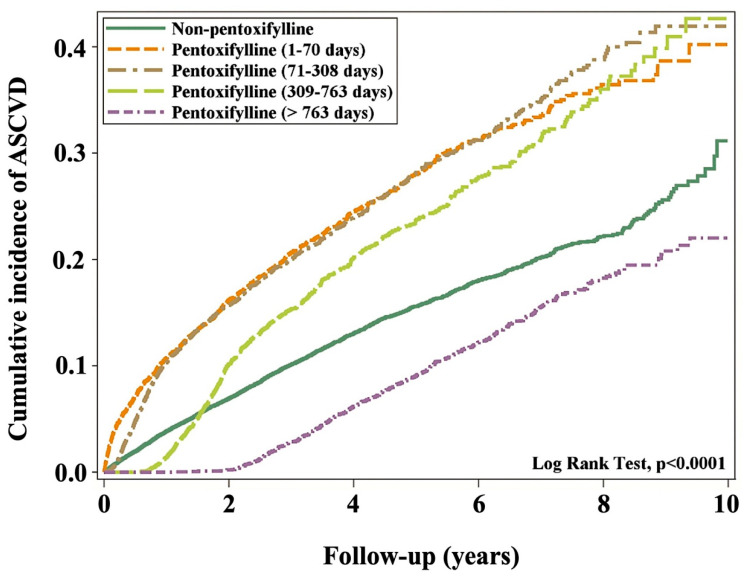
Cumulative incidence of ASCVD and use of pentoxifylline medication. ASCVD, atherosclerotic cardiovascular disease.

**Figure 3 antioxidants-13-01471-f003:**
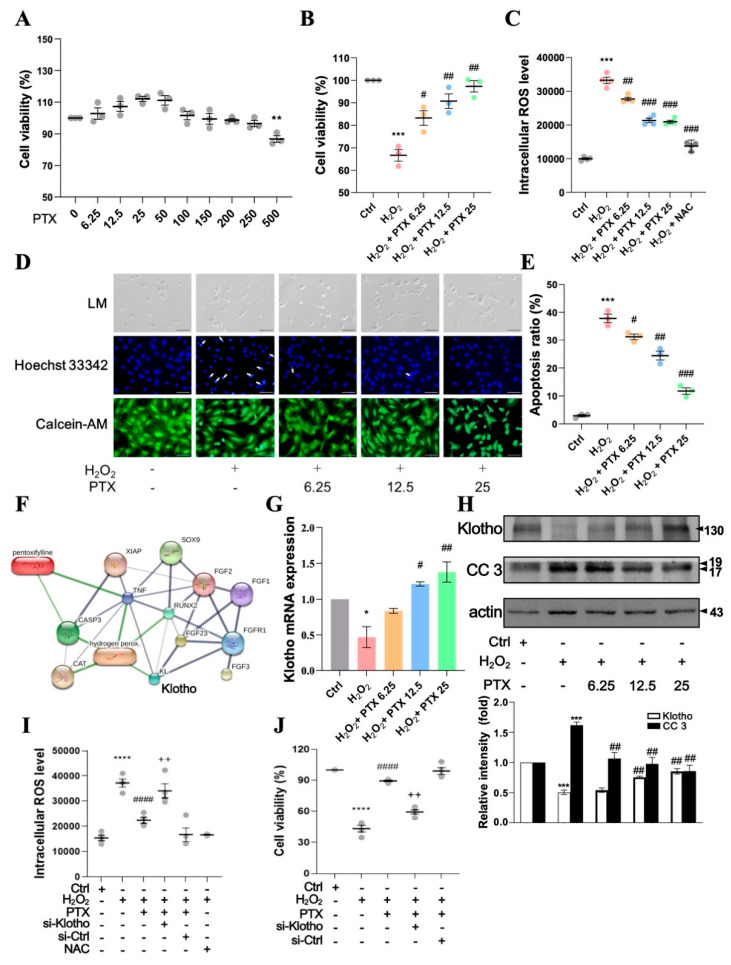
Pentoxifylline reduces endothelial cell apoptosis in a dose-dependent manner. (**A**) Viability of PTX (0–500 μM)-treated endothelial cells for 24 h based on the WST-1 assay. (**B**) PTX protects HAECs treated with H_2_O_2_. Viability of endothelial cells treated with H_2_O_2_ (100 µM) and PTX based on WST-1 assay. (**C**) Level of intracellular ROS measured using DCFH_2_-DA staining. (**D**) Hoechst 33342 and Calcein-AM staining showing cell apoptosis. (Scale bar = 10 μm.) White arrows: HAECs with condensed, fragmented nuclei were considered to be undergoing apoptosis. (**E**) Quantification of apoptotic cells. (**F**) PTX facilitates protein–protein interactions. The line size is proportional to the combined fraction of interactions. (**G**) mRNA level of Klotho. (**H**) Protein levels of Klotho, cleaved-caspase 3 (CC3), and β-actin. HAECs were exposed to PTX (6.25, 12.5, and 25 μM) before H_2_O_2_ (100 μM) treatment for 24 h. (**I**) intracellular ROS levels were measured. (**J**) Cell viability. HAECs pretreated with si-Klotho with or without PTX (25 μM) prior to a 24 h treatment with H_2_O_2_ (100 μM). Data are presented as mean ± SD (*n* = 3). The *p*-values were determined using a Student’s *t*-test. * *p* < 0.05, ** *p* < 0.01, *** *p* < 0.001, **** *p* < 0.0001 vs. control group or 0 μM group; # *p* < 0.05, ## *p* < 0.01, ### *p* < 0.001, #### *p* < 0.0001 vs. H_2_O_2_ group; ^++^ *p* < 0.01 vs. PTX + H_2_O_2_ group. ROS: reactive oxygen species; PTX: pentoxifylline; ctrl; control; NAC: N-acetyl cysteine; SD: standard deviation; Calcein-AM: calcein acetoxymethyl ester; HAECs: human aortic endothelial cells; DCFH_2_-DA: 2′,7′-dichlorofluorescein diacetate; WST-1: water-soluble tetrazolium salt; LM: light microscopy.

**Figure 4 antioxidants-13-01471-f004:**
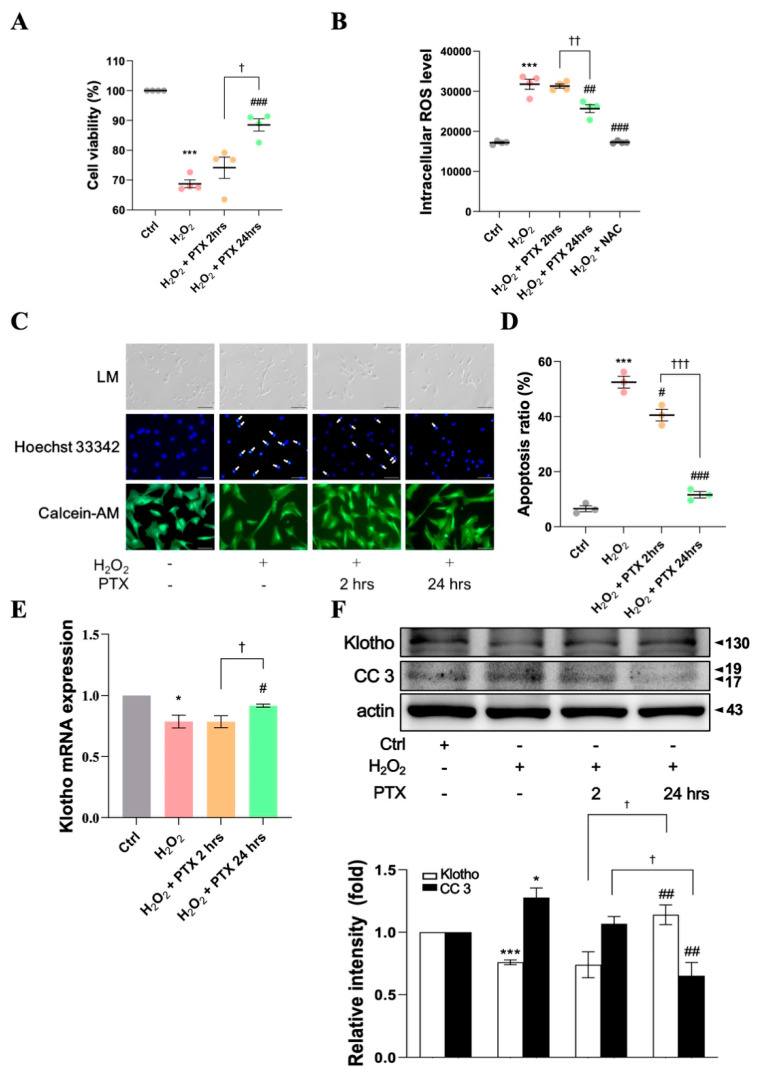
Protective effects of short- or long-term PTX exposure on H_2_O_2_-induced endothelial cell apoptosis. HAECs were exposed to PTX (25 μM) for 2 h or 24 h before H_2_O_2_ (100 μM) treatment for 24 h. (**A**) Viability of H_2_O_2_ (100 μM) and PTX (25 μM)-treated endothelial cells based on the WST-1 assay. (**B**) Intracellular ROS levels in H_2_O_2_ and PTX-treated cells (2 h or 24 h) were measured using DCFH_2_-DA staining. N-acetyl cysteine (NAC, 5 mM) as an antioxidant [29]. (**C**) Hoechst 33342 and Calcein-AM staining showed cell apoptosis. White arrows: HAECs with condensed, fragmented nuclei were considered to be undergoing apoptosis. (**D**) Quantification of apoptotic cells. (**E**) mRNA level of Klotho. (**F**) Protein levels of Klotho, cleaved-caspase 3, and β-actin. Data are expressed as mean ± SD (*n* = 3–4). The *p*-values were determined using a Student’s *t*-test. * *p* < 0.05, *** *p* < 0.001 vs. control or 0 µM group; # *p* < 0.05, ## *p* < 0.01, ### *p* < 0.001 vs. H_2_O_2_ group. ^†^ *p* < 0.05, ^††^ *p* < 0.01, ^†††^ *p* < 0.001 vs. H_2_O_2_ + PTX 2 h group. PTX: pentoxifylline; ctrl; control; NAC: N-acetyl cysteine; SD: standard deviation; HAECs: human aortic endothelial cells; Calcein-AM: calcein acetoxymethyl ester; DCFH_2_-DA, 2′,7′-dichlorofluorescein diacetate; WST-1, water-soluble tetrazolium salt; LM: light microscopy.

**Figure 5 antioxidants-13-01471-f005:**
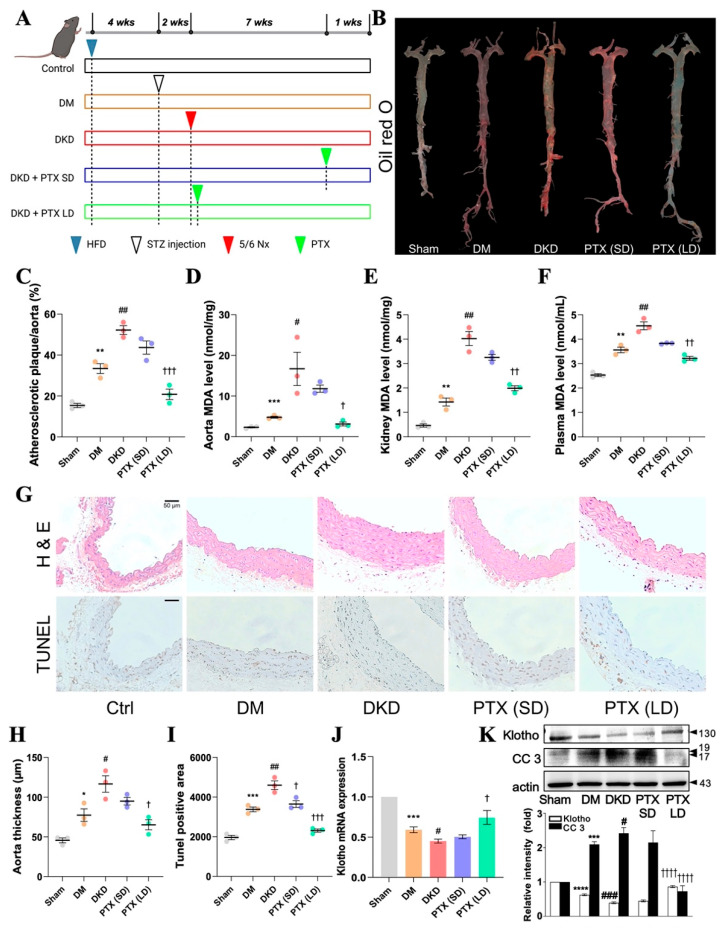
Pentoxifylline decreases plaque formation and cell apoptosis in DKD mice. (**A**) Schematic illustration of the experimental mouse model. (**B**) Representative images of the aorta from mice in each group stained with Oil Red O. (**C**) Quantification of the aortic root lesion sizes. (**D**) MDA levels in the aorta. (**E**) MDA levels in the kidneys. (**F**) MDA levels in plasma. (**G**) Representative images of the aorta from mice in each group stained with H&E and TUNEL (Scale bar = 50 μm). (**H**) Quantification of aorta thickness. (**I**) Quantification of TUNEL-positive cells in the aorta. (**J**) mRNA levels of Klotho. (**K**) Protein levels of Klotho, cleaved-caspase 3, and β-actin in the aorta of experimental mice. Data are expressed as mean ± SD (*n* = 3). * *p* < 0.05, ** *p* < 0.01, *** *p* < 0.001, **** *p* < 0.0001 vs. sham group; # *p* < 0.05, ## *p* < 0.01, ### *p* < 0.001 vs. DM group; ^†^ *p* < 0.05, ^††^ *p* < 0.01, ^†††^ *p* < 0.001, ^††††^ *p* < 0.0001 vs. DKD group. DM group: STZ-induced mice; DKD group: 5/6Nx in STZ-induced mice. DKD, diabetic kidney disease; MDA, malondialdehyde; H&E, hematoxylin and eosin; TUNEL, terminal deoxynucleotidyl transferase-mediated dUTP nick end labeling; STZ, streptozotocin; SD: standard deviation; PTX, pentoxifylline; Ctrl, control; 5/6 Nx, 5/6 nephrectomy; DM, diabetes mellitus; SD, standard deviation.

**Table 1 antioxidants-13-01471-t001:** Characteristics of the study population according to pentoxifylline stratification.

Variable	Non-Pentoxifylline Users	Pentoxifylline Users	*p*-Value
	*n* (%)/Mean ± SD	*n* (%)/Mean ± SD	
**All**	31,141	31,141	
**Sex**			0.9232
Female	14,529 (46.66)	14,541 (46.69)	
Male	16,612 (53.34)	16,600 (53.31)	
**Age (y)**			0.9992
<50	2642 (8.48)	2649 (8.51)	
50–59	5508 (17.69)	5491 (17.63)	
60–69	8536 (27.41)	8560 (27.49)	
70–79	8186 (26.29)	8171 (26.24)	
≥80	6269 (20.13)	6270 (20.13)	
Mean age	67.75 ± 13.00	67.75 ± 12.96	0.9914
**Comorbidities**			
AMI			0.2974
No	30,884 (99.17)	30,907 (99.25)	
Yes	257 (0.83)	234 (0.75)	
Hypertension			0.8354
No	4352 (13.98)	4370 (14.03)	
Yes	26,789 (86.02)	26,771 (85.97)	
COPD			0.7386
No	26,352 (84.62)	26,382 (84.72)	
Yes	4789 (15.38)	4759 (15.28)	
LC			0.5868
No	29,631 (95.15)	29,660 (95.24)	
Yes	1510 (4.85)	1481 (4.76)	
Autoimmune diseases			0.2023
No	30,884 (99.17)	30,912 (99.26)	
Yes	257 (0.83)	229 (0.74)	
**Follow-up period (y)**	3.09 ± 2.13	3.18 ± 2.20	<0.0001

SD, standard deviation; AMI, acute myocardial infarction; COPD, chronic obstructive pulmonary disease; LC, liver cirrhosis.

**Table 2 antioxidants-13-01471-t002:** Incidence and HR of ASCVD between the pentoxifylline cohorts and characteristics of patients with DKD.

Variable	Event	Person-y	IR	Crude		Adjusted	
	N = 8134		100 Person-y	HR (95% CI)	*p*-Value	HR (95% CI)	*p*-Value
**Pentoxifylline**							
No	3321	96,190	3.45	1 (Reference)		1 (Reference)	
Yes	4813	99,149	4.85	1.41 (1.35, 1.47)	<0.0001	1.41 (1.34, 1.47)	<0.0001
**Sex**							
Female	3587	91,169	3.93	1 (Reference)		1 (Reference)	
Male	4547	104,170	4.36	1.11 (1.06, 1.16)	<0.0001	1.12 (1.07, 1.17)	<0.0001
**Age (y)**							
<50	672	19,274	3.49	1 (Reference)		1 (Reference)	
50–59	1585	38,989	4.07	1.16 (1.06, 1.27)	0.0010	1.14 (1.04, 1.24)	0.0058
60–69	2385	55,353	4.31	1.22 (1.12, 1.33)	<0.0001	1.16 (1.07, 1.27)	0.0007
70–79	2213	50,785	4.36	1.24 (1.13, 1.35)	<0.0001	1.14 (1.04, 1.24)	0.0042
≥80	1279	30,938	4.13	1.15 (1.05, 1.27)	0.0030	1.04 (0.94, 1.14)	0.4763
Mean age							
**Comorbidities**							
AMI							
No	8013	194,346	4.12	1 (Reference)		1 (Reference)	
Yes	121	993	12.18	2.88 (2.41, 3.44)	<0.0001	2.75 (2.30, 3.30)	<0.0001
Hypertension							
No	792	30,352	2.61	1 (Reference)		1 (Reference)	
Yes	7342	164,987	4.45	1.69 (1.57, 1.82)	<0.0001	1.66 (1.54, 1.79)	<0.0001
COPD							
No	6909	169,267	4.08	1 (Reference)		1 (Reference)	
Yes	1225	26,072	4.70	1.14 (1.07, 1.21)	<0.0001	1.10 (1.03, 1.17)	0.0025
LC							
No	7866	187,425	4.20	1 (Reference)		1 (Reference)	
Yes	268	7913	3.39	0.80 (0.71, 0.90)	0.0003	0.79 (0.70, 0.90)	0.0002
Autoimmune diseases							
No	8065	193,904	4.16	1 (Reference)		1 (Reference)	
Yes	69	1435	4.81	1.15 (0.91, 1.46)	0.2445	1.19 (0.94, 1.50)	0.1588

HR, hazard ratio; ASCVD, atherosclerotic cardiovascular disease; DKD, diabetic kidney disease; COPD, chronic obstructive pulmonary disease; CI, confidence interval; LC, liver cirrhosis; AMI, acute myocardial infarction; IR, incidence rate.

**Table 3 antioxidants-13-01471-t003:** ASCVD risks by the duration of pentoxifylline treatment.

Variable	Event	Person-y	IR	Crude		Adjusted	
	N = 8134		100 Person-y	HR (95% CI)	*p*-Value	HR (95% CI)	*p*-Value
**Pentoxifylline**							
No	3321	96,190	3.45	1 (Reference)		1 (Reference)	
1–70 days	1546	20,890	7.40	2.14 (2.01, 2.27)	<0.0001	2.16 (2.03, 2.29)	<0.0001
71–308 days	1578	21,636	7.29	2.10 (1.98, 2.23)	<0.0001	2.11 (1.99, 2.24)	<0.0001
309–763 days	1040	20,932	4.97	1.42 (1.33, 1.53)	<0.0001	1.41 (1.32, 1.51)	<0.0001
>763 days	649	35,690	1.82	0.53 (0.49, 0.58)	<0.0001	0.53 (0.49, 0.58)	<0.0001

ASCVD, atherosclerotic cardiovascular disease; CI, confidence interval; HR, hazard ratio; IR: incidence rate.

## Data Availability

Owing to existing data sharing permissions, summary statistics are available on request to the authors.

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
