# Peer review of "Long-Term Pentoxifylline Therapy Is Associated with a Reduced Risk of Atherosclerotic Cardiovascular Disease by Inhibiting Oxidative Stress and Cell Apoptosis in Diabetic Kidney Disease Patients"

_antioxidants, 2024, doi:10.3390/antiox13121471_

Round 1
Reviewer 1 Report
In the study “Long-term pentoxifylline therapy reduces atherosclerotic cardiovascular disease risk by inhibiting reactive oxygen species production and cell apoptosis in patients with diabetic kidney disease”, the authors explored the benefit of pentoxifylline treatment on atherosclerotic events in the context of diabetic kidney disease. The study is well-designed and clear, however, the mechanistic insights are slightly poor with room to be improved.
Major suggestions:
The authors described an increase of intracellular ROS with the very unspecific DCF-DA probe. Considering the scope of this journal, it would be necessary to analyze the nature of these ROS a bit deeper and provide additional ROS measures and/or ROS-induced damage measures, such as MDA or 8-hydroxy-uracyl determination. In the same line, to explore the sources of ROS is important.
Experiments with antioxidants should be carried out in cells and mice to analyze whether they produce a similar effect as PTX.
To prove the Klotho-mediated PTX effect, experiments in cells with inhibitors/siRNA should be performed.
In both, mice and cell culture models, the authors found a better effect of long treatment with PTX than with short treatment, however, the short treatment still has a beneficial effect. How do authors explain the increase of the risk in patients with short PTX treatment? Please add a paragraph to the discussion.
The article “doi: 10.1186/s12933-024-02393-x” should be cited and discussed.
In the discussion, the authors state “Pentoxifylline is a potent antioxidant that inhibits ROS generation by activating antioxidant enzymes” (lines 163-164), however, the protective role of PTX is attributed to a klotho upregulation. Please explore the expression of the antioxidant defenses in both the mice and the cell culture model.
Reviewer 2 Report
This is an interesting study which aimed to evaluate the effects of pentoxifylline (PTX) on cardiovascular outcomes in patients with diabetic chronic kidney disease (CKD) without established cardiovascular disease (CVD) and to elucidate the protective mechanisms of PTX in vascular endothelial cells (L21-24). The final analysis utilized data from Taiwan's Health and Welfare Data Science Center, encompassing a large cohort of 62,282 participants over an 11-year period. The extensive dataset, combined with a comprehensive analysis using a multi-model approach, enhances the study's robustness and its generalizability. Overall, this paper has the potential to make a valuable contribution to the field. However, the authors should revise the manuscript to address the following issues before further consideration. Line numbers are provided for reference where applicable.
Major issues:
a) In the animal study, short-term (SD) and long-term (LD) PTX treatments were defined as 8 weeks and 12 weeks, respectively (L217-218). For the human study, based on the quartiles of treatment durations, the authors arbitrarily set a cut-off of >763 days (approximately two years) to define "long-term" PTX therapy without providing a clear rationale or references for this threshold. This raises concerns regarding the comparability of treatment durations across species. Furthermore, the authors present mechanisms observed in the animal study, such as reductions in ROS and apoptosis, as relevant to the human study findings. While these mechanisms are likely relevant to ASCVD and DKD, directly applying findings from mice to human populations with a much longer treatment duration introduces uncertainty. It is unclear whether the 12-week duration in mice appropriately mirrors the effects of long-term PTX therapy over 763 days in humans.
b) The observed trend in the Results (Table 3), where hazard ratios progressively decrease with longer PTX use and only become protective (HR < 1) after an extended period (>763 days), is intriguing. Such substantial cumulative effects emerging only after two years are uncommon, especially given the lack of prior evidence indicating that PTX exhibits such a long latency before reaching a protective threshold. In contrast, the figures showing in vitro and animal model data suggest that PTX may have an immediate impact on cellular mechanisms relevant to ASCVD, such as reducing ROS production and apoptosis. This discrepancy indicates that, while PTX may contribute to reducing ASCVD risk, the prolonged duration required to observe a protective effect in humans is not clearly corroborated by the in vitro and animal model data. The authors should discuss these findings in light of studies reporting similarly delayed impacts of PTX on chronic conditions, if such references are available, to better contextualize the extended time frame observed here.
c) In the manuscript, Line 144 states that “Our cell studies were conducted with primary human aortic ECs (HAECs)...” However, under Materials and Methods, there is no detailed characterization of the human subjects from whom these endothelial cells were derived. Without clear information on the origin of these HAECs, it is uncertain whether they accurately represent the endothelial cells of diabetic kidney disease (DKD) patients with various comorbidities included in this study. Such patients may experience endothelial dysfunction and other cellular changes specific to their condition, which could significantly influence cellular responses to PTX. This information is crucial for assessing the applicability of the findings to the current study population and for understanding the limitations of the cellular model used. In fact, without a clear bridge between the animal model findings and human treatment conditions, there is a risk of overgeneralizing the results.
[Including all the above comments]
This is an interesting study which aimed to evaluate the effects of pentoxifylline (PTX) on cardiovascular outcomes in patients with diabetic chronic kidney disease (CKD) without established cardiovascular disease (CVD) and to elucidate the protective mechanisms of PTX in vascular endothelial cells (L21-24). The final analysis utilized data from Taiwan's Health and Welfare Data Science Center, encompassing a large cohort of 62,282 participants over an 11-year period. The extensive dataset, combined with a comprehensive analysis using a multi-model approach, enhances the study's robustness and its generalizability. Overall, this paper has the potential to make a valuable contribution to the field. However, the authors should revise the manuscript to address the following issues before further consideration. Line numbers are provided for reference where applicable.
Major issues:
a) In the animal study, short-term (SD) and long-term (LD) PTX treatments were defined as 8 weeks and 12 weeks, respectively (L217-218). For the human study, based on the quartiles of treatment durations, the authors arbitrarily set a cut-off of >763 days (approximately two years) to define "long-term" PTX therapy without providing a clear rationale or references for this threshold. This raises concerns regarding the comparability of treatment durations across species. Furthermore, the authors present mechanisms observed in the animal study, such as reductions in ROS and apoptosis, as relevant to the human study findings. While these mechanisms are likely relevant to ASCVD and DKD, directly applying findings from mice to human populations with a much longer treatment duration introduces uncertainty. It is unclear whether the 12-week duration in mice appropriately mirrors the effects of long-term PTX therapy over 763 days in humans.
b) The observed trend in the Results (Table 3), where hazard ratios progressively decrease with longer PTX use and only become protective (HR < 1) after an extended period (>763 days), is intriguing. Such substantial cumulative effects emerging only after two years are uncommon, especially given the lack of prior evidence indicating that PTX exhibits such a long latency before reaching a protective threshold. In contrast, the figures showing in vitro and animal model data suggest that PTX may have an immediate impact on cellular mechanisms relevant to ASCVD, such as reducing ROS production and apoptosis. This discrepancy indicates that, while PTX may contribute to reducing ASCVD risk, the prolonged duration required to observe a protective effect in humans is not clearly corroborated by the in vitro and animal model data. The authors should discuss these findings in light of studies reporting similarly delayed impacts of PTX on chronic conditions, if such references are available, to better contextualize the extended time frame observed here.
c) In the manuscript, Line 144 states that “Our cell studies were conducted with primary human aortic ECs (HAECs)...” However, under Materials and Methods, there is no detailed characterization of the human subjects from whom these endothelial cells were derived. Without clear information on the origin of these HAECs, it is uncertain whether they accurately represent the endothelial cells of diabetic kidney disease (DKD) patients with various comorbidities included in this study. Such patients may experience endothelial dysfunction and other cellular changes specific to their condition, which could significantly influence cellular responses to PTX. This information is crucial for assessing the applicability of the findings to the current study population and for understanding the limitations of the cellular model used. In fact, without a clear bridge between the animal model findings and human treatment conditions, there is a risk of overgeneralizing the results.
Other issues:
Title:
a) The title is relatively long and could be improved by condensing or streamlining some terms to broader concepts for clarity.
b) The title asserts that "long-term pentoxifylline therapy reduces atherosclerotic cardiovascular disease risk," which may be an overstatement. Although the authors explored mechanistic pathways using in vitro and animal models, extrapolating these findings directly to the human cohort may be speculative. A more cautious phrasing, such as "...is associated with a reduced risk of..." would better reflect the observational nature of the study.
Abstract:
a) L20-21: The sentence, "Little is known about which pentoxifylline (PTX) regimen provides optimal atherosclerotic cardiovascular disease (ASCVD) outcomes," is quite vague and could be more specific in describing the study's focus.
b) L21-24: The aim of this study, "to assess the effects of PTX on cardiovascular outcomes in cases of diabetic chronic kidney disease without established cardiovascular disease," is broad and lacks specificity. Please consider rephrasing to provide a more targeted and precise objective.
c) L25-26: The abstract refers to a sample size from a "nationwide cohort" of "23,000,000 participants," which may give the impression that all 23 million participants were included in the analysis. Please specify and clarify the actual sample size of diabetic kidney disease (DKD) patients used in this study.
d) L27-28 and L31-32: Please define “short-term” and “long-term” PTX treatment early in the manuscript by briefly specifying the cut-off point between them.
e) L32-35: The text used here might be confusing, as it does not clearly distinguish between findings from patient data and those from cell culture experiments.
Introduction:
a) The introduction lacks specific details on what remains inadequately explored about the role of pentoxifylline (PTX) in the context of ASCVD and DKD. Additionally, the authors do not provide a thorough summary of the existing body of knowledge or explicitly highlight the limitations and gaps in the current literature regarding the effects of PTX, especially long-term, on ASCVD in patients with DKD.
b) Although the authors mention that PTX has anti-inflammatory and renoprotective properties, they do not clearly explain why these attributes might make PTX particularly advantageous for treating DKD patients at risk of ASCVD. Furthermore, a brief discussion of the limitations of other therapeutic agents for ASCVD prevention in DKD patients would help clarify the unique perspective and value of this research.
c) The rationale for comparing long-term versus short-term PTX therapy in the context of DKD and ASCVD is not adequately justified.
d) The introduction does not clearly and adequately explain Klotho’s relationship with ASCVD and DKD, or why it is relevant to the study. Providing a clearer concise explanation of the relevance of mechanistic aspects, such as ROS and Klotho protein, in reducing ASCVD risk would help bridge the gap between the clinical outcomes and underlying cellular/molecular mechanisms.
e) Discussing how both the clinical and mechanistic effects of PTX could lead to more targeted treatments would better justify the study’s approach and help readers understand how this research could improve treatment strategies.
f) Please ensure consistent terminology throughout the whole text (including the title), particularly with terms like ASCVD versus cardiovascular disease, to maintain clarity and precision.
Materials and Methods:
Main issues have been pointed out under ‘Major issues’ above.
Others-
a) While the study includes DKD patients without pre-existing ASCVD, it does not provide clear criteria for diagnosing DKD. Defining these criteria would improve clarity and reproducibility.
b) Although dosing details are provided for the cell and animal studies, there is no justification regarding their physiological relevance to human dosing regimens. The authors should include a brief explanation or references to support the chosen doses and discuss their comparability to human treatments.
c) In the statistical analysis, the variables considered for adjustment across different models should be clearly specified.
Results:
a) The authors aimed to assess the effect of PTX on ASCVD risk in diabetic kidney disease (DKD) patients and explore the cellular mechanisms underlying PTX's action. However, the Results section does not clearly and systematically present how each finding aligns with these specific aims. Additionally, it lacks a clear distinction between results derived from the cohort study and those from in vitro and animal models. This blending of epidemiological data with experimental findings may confuse readers about which results are applicable to human patients versus those observed under controlled experimental conditions.
b) The authors mentioned a "47% reduction" in ASCVD incidence among long-term PTX users (L29), which is a significant outcome of the study. However, this result is not explicitly presented in the Results section. Including specific statistics, such as confidence intervals and p-values, would enhance the clarity and reliability of this finding. Providing these details would help readers better understand the significance of long-term PTX therapy on ASCVD risk.
c) The term "ASCVD goals" is unclear. The authors should clarify what is meant by this phrase. Additionally, the wording “failed to achieve them in the following years” is vague and would benefit from greater specificity.
d) While the tables and figures are presented with titles and legends, they often lack comprehensive descriptions. To improve clarity and interpretability, it is important to ensure that all tables and figures are adequately labeled with detailed information, including sample sizes, concentrations, and specific adjustments made in statistical models. For instance, some tables do not specify which covariates were adjusted for in the models, which is critical for assessing the robustness of the results.
Discussion:
a) The Discussion section addresses the observed results but does not sufficiently link them to the study's title and specific objectives. While the findings on ASCVD reduction and cellular mechanisms are mentioned, they are not well contextualized within the study’s aim to compare the effects of long-term versus short-term PTX therapy.
b) Although the paper mentions a 47% reduction in ASCVD risk among long-term PTX users, the interpretation lacks depth. There should be a more thorough exploration of how these findings align with or differ from existing literature on PTX, especially concerning its long-term efficacy. Comparing these results with other therapies commonly used for ASCVD prevention in DKD patients could also provide useful context for readers.
c) The discussion of mechanistic insights from in vitro and animal studies does not clearly connect to the epidemiological findings. While the authors reference pathways involving ROS and Klotho, they do not explain how these mechanisms might lead to the delayed protective effects seen in human patients. This gap makes it challenging to fully assess the relevance of these mechanisms to the clinical outcomes.
d) The authors should consider condensing the current text and removing redundancies, particularly in sections discussing PTX. This would improve the readability and flow of the manuscript.
e) The current structure of the discussion alternates between clinical and cellular findings without clear transitions. Organizing the Discussion clearly into sections for clinical results, mechanistic insights, and limitations would create a more logical structure and make it easier for readers to follow the arguments.
f) The limitations section does not fully address key issues and could be improved by directly discussing how these limitations may impact the study’s findings and the generalizability of its conclusions. For instance, using liver disease and COPD as proxies for BMI, alcohol intake, and smoking is insufficient, as these indirect measures may not fully capture the effects of these factors on ASCVD risk. Additionally, the absence of lipid profile data—essential for assessing cardiovascular risk—significantly hampers the study's ability to account for critical ASCVD risk factors and warrants further emphasis. While the authors note demographic consistency with census data, reliance on claims data without clinical details may introduce biases and restrict generalizability.
g) The Conclusion would be more impactful if it highlighted key findings (such as the information on 47% reduction in ASCVD risk) to underscore the importance of the findings.
Minor point:
Please define all abbreviations upon their first mention
Reviewer 3 Report
The manuscript presents an exciting paper on long-term pentoxifylline therapy, which reduces the risk of atherosclerotic cardiovascular disease by inhibiting reactive oxygen species production and cell apoptosis in patients with diabetic kidney disease. The paper consists of 3 parts: statistical studies, cell line studies, and a mouse model indicating careful consideration of the topic.
The research highlights how this therapy may inhibit the production of reactive oxygen species and reduce cell apoptosis, paving the way for better cardiovascular health outcomes.
The study demonstrated that pentoxifylline has protective effects against oxidative stress and apoptosis, which are critical factors in the progression of cardiovascular diseases.
The study showed that Pentoxifylline inhibited the production of reactive oxygen species (ROS). The methodology involved measuring intracellular ROS levels using a specific assay kit, which indicated that pentoxifylline treatment led to a decrease in ROS levels compared to controls. This reduction in oxidative stress is significant as high levels of ROS are associated with increased cell apoptosis and cardiovascular complications.
The findings imply that pentoxifylline could serve as a therapeutic agent to mitigate cardiovascular risks associated with diabetic kidney disease. By reducing oxidative stress and apoptosis, pentoxifylline may improve overall cardiovascular health in these patients. This suggests a potential shift in treatment strategies, emphasizing the importance of managing oxidative stress in the prevention of cardiovascular diseases in individuals with diabetic kidney disease.
However, before the paper is published some of the issues must be clarified as listed in detailed comments.
Before the paper is published some of the issues must be clarified
1. The introduction needs a bit of an update and justification of the studies.
2. Line 43: find more recent data.
Examples - China: The Writing Committee of the Report on Cardiovascular Health and Diseases in China. Report on Cardiovascular Health and Diseases in China 2021: An Updated Summary[J]. Biomedical and Environmental Sciences, 2022, 35(7): 573-603. doi: 10.3967/bes2022.079
Bi L, et al. (2022) Atherosclerotic Cardiovascular Disease Risk and Lipid-Lowering Therapy Requirement in China. Front. Cardiovasc. Med. 9:839571. doi: 10.3389/fcvm.2022.839571
The Writing Committee of the Report on Cardiovascular Health and Diseases in China. Report on Cardiovascular Health and Diseases in China 2021: An Updated Summary[J]. Biomedical and Environmental Sciences, 2022, 35(7): 573-603. doi: 10.3967/bes2022.079
Examples worldwide: https://www.acc.org/About-ACC/Press-Releases/2023/12/11/18/48/New-Study-Reveals-Latest-Data-on-Global-Burden-of-Cardiovascular-Disease
3. State clearly the number of data 129,764 and not 23,000,000 in Introduction.
4. Tables 1-3 – please format to the Antioxidants standards. Explain in Materials and Methods what Adjusted variable means
5. Page 11 lines 5 -10 what does it mean that “participants using PTX for 1-70d could not achieve their ASCVD goals” please define what the goal means here
6. Shorten the title to: Long-term pentoxifylline therapy reduces the risk of atherosclerotic cardiovascular disease by inhibiting oxidative stress and cell apoptosis in diabetic kidney disease patients.
7. Page 5 line 179 H2O2
8. Unify style of Materials and Methods
we applied, used – was applied, used etc.
9. Increase the resolution of Figure 2
Round 2
Reviewer 1 Report
The authors addressed all my comments
The authors addressed all my comments
Reviewer 2 Report
The authors have addressed the critical points raised in the initial review. I have no further comments or concerns at this time.
No further comments or concerns at this time.